# Design 2-Speed Transmission for Compact Electric Vehicle Using Dual Brake System

**Jae-Oh Han, Jae-Won Shin, Jae-Chang Kim and Se-Hoon Oh ***

College of Engineering, Chung-ang University, 84, Heukseok-ro, Dongjak-gu, Seoul 06974, Korea;
laioh@cau.ac.kr (J.-O.H.); shinjwgo@naver.com (J.-W.S.); zc0607@cau.ac.kr (J.-C.K.)

* Correspondences: osh@cau.ac.kr; Tel.: +82-10-9172-3401

**Abstract:** Mega trends in the global automotive industry are environmentally friendly. As laws and regulations tighten at the government level, the automobile industry is striving to develop a drive system that can operate without using fossil fuels, instead of developing an internal combustion engine using fossil fuels. Environmentally-friendly energy is attracting attention as an alternative to solve the problems of air pollution and fossil fuel depletion. Electricity is attracting the most attention among environmentally-friendly alternative fuels. In addition, research on the development of a high-efficiency and high-reliability advanced electric automobile drive system are actively being carried out. In this study, a two-speed transmission for electric vehicles is developed using environmentally-friendly fuel. The 1st and the 2nd planetary gear modules were integrated, the ring gear and the carrier gear were shared, and the dual disc brake was used to design a mechanism for fixing each sun and shifting gear. Such a structure can improve shift energy efficiency compared to that of conventional transmissions. It was judged that the structure was suitable for an electric car using a limited power supply. Each gear was designed by calculating bending strength and surface durability.

**Keywords:** environmentally friendly; downsizing; fuel efficiency; electric vehicle; planetary gear; transmission

## 1. Introduction

The global automotive industry is actively working to reduce weight by downsizing and developing lighter components to increase fuel efficiency and reduce emissions [1–6]. If the weight of a vehicle is reduced by 10%, fuel efficiency is improved by about 3 to 8%, acceleration performance (0~100 Km reach time) is improved by 8%, and the emission of environmental pollutants is also reduced (carbon monoxide 4.5%, NOx 8.8%, hydrocarbons 2.5%). In addition, the performance of the steering wheel is improved by approximately 6%, the durable life of the chassis of the automobile is improved by approximately 1.7 times, and the braking stop distance is shortened by 5% [7]. Governments around the world are emphasizing 'fuel efficiency improvement' and 'environmentally friendly', which are key to changes in the future automobile industry, by strengthening environmental regulations to cope with climate change [8–11]. The spread of electric vehicles using environmentally-friendly fuels is increasing. An electric car uses a motor and controls the speed by controlling the motor output. So, the usual electric vehicle uses a reducer. However, it does not matter when the load is small, such as flat running, but it is stable to use the transmission to control the motor output when the load is large, such as when going up a hill. There are many high and low hills in Korea, as well as in most cities around the world; a lot of roads require high torque. There is a need for a transmission that is adaptable to various driving environments and that optimizes torque and speed according to the driving speed [12].

## 2. Design 2-Speed Transmission Development

### 2.1. The Purpose of the Study

Most cities around the world have large and small hills and unpaved roads, and it takes a lot of torque to travel these roads. In order to cope with such varied driving environments, a transmission that optimizes the torque and the speed according to the running speed is required; it is therefore necessary for an electric car with a low output relative to the weight of the vehicle. In this study, we developed a transmission optimized for an electric motor of 88 KW with a maximum torque of 300 Nm when the motor rpm is 2850. A two-speed transmission system was planned to use a planetary gear set and its structure was designed. Also, the gear train was designed based on the calculated gear strength.

### 2.2. Conception of 2-Speed Transmission Development

To design a transmission concept using a planetary gear set, and to design more compact than the conventional one, the first and second planetary gear sets were arranged in series, instead of parallel, to reduce unnecessary space. The first and second planetary gear sets share a ring gear and a carrier. By removing the duplicate parts, it was possible to create a more compact design.

Figure 1 shows the design theory. A planetary gear set for the first stage and a planetary gear set for the second stage are integrally formed. Z1 is a two-speed sun gear, Z2 is a two-speed planetary gear, Z3 is a one-stage planetary gear, Z4 is a single-speed sun gear, and Z5 is a ring gear. The first sun gear consists of a hollow shaft, with gears on the outside of the hollow shaft. The second sun gear is a solid shaft, gearing out of the solid shaft. The first and second planetary gears are each three, and the first and second planetary gears are integrally formed. 1-B is a more concrete picture of design theory. Z_S1 is a single-stage sun gear, Z_S2 is a two-speed sun gear, Z_P1 is a one-stage planetary gear, Z_P2 is a two-stage planetary gear, and Z_R is a ring gear.

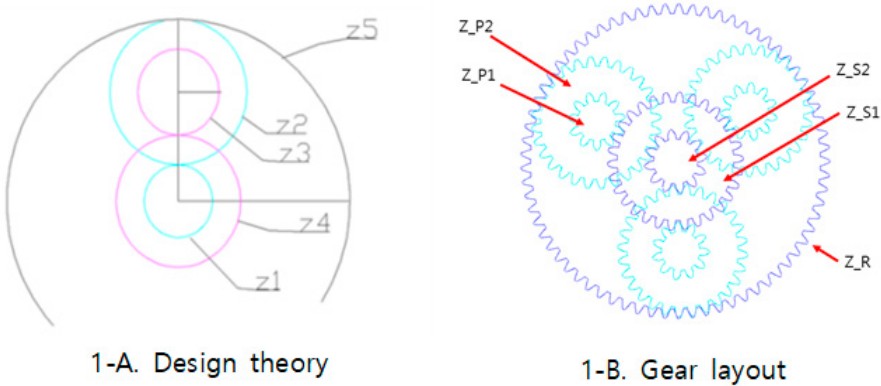

**Figure 1.** Concept drawing.

Figure 2 illustrates the existing transmission configuration and the new type of 2-speed transmission concept.

### 2.3. Design of 2-Speed Transmission

#### 2.3.1. 3D Modeling New Type 2-Speed Transmission

Figures 3 and 4 are a 3D model of a new type of two-speed transmission. When the power is transmitted from the motor to the planetary gear unit, the power is transmitted to the differential gear through the carrier of the planetary gear unit.

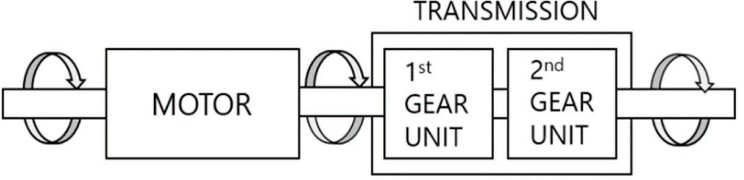

2-A **1st gear unit and 2nd gear unit are separated**

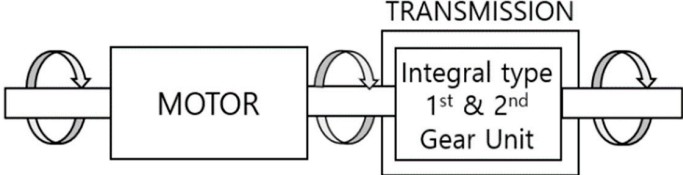

**2-B** 1st gear unit and 2nd gear unit are integrated

**Figure 2.** Concepts of space separation of 1st gear unit and 2nd gear unit.

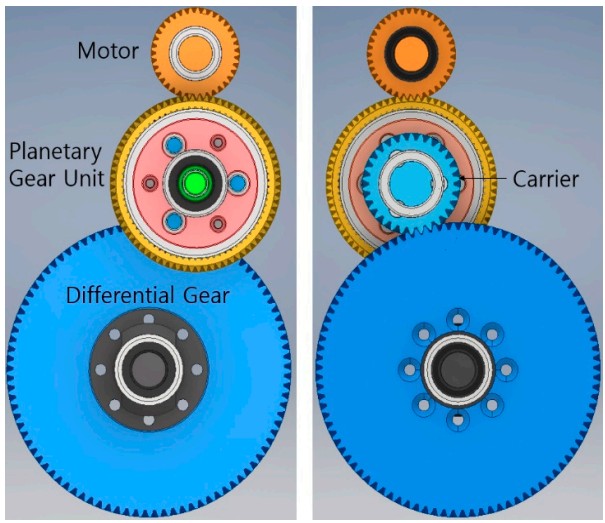

**Figure 3.** New type 2-speed transmission 3D modeling.

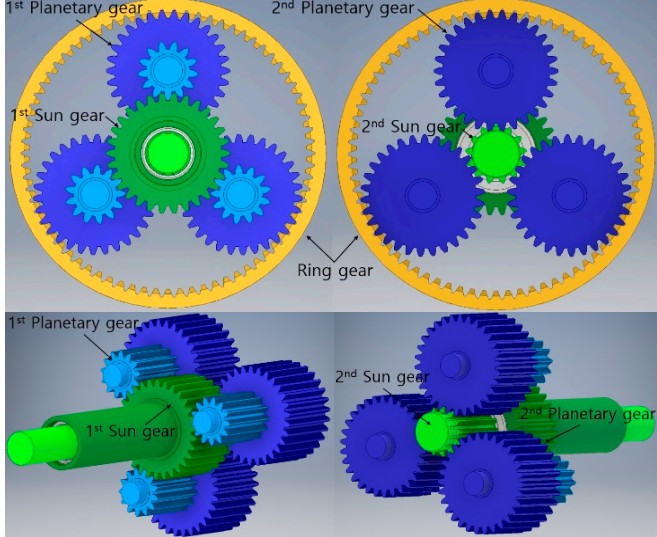

**Figure 4.** The internal structure of the integral type planetary gear unit.

Sun gear, ring gear and carrier rotate about the same axis, and planetary gears are assembled to the carrier. Depending on which parts are to be fixed and setting them as input shaft or output shaft, the gear ratio and the direction of rotation are determined. To satisfy the required transmission ratio, the ring gear is fixed, setting the sun gear to the input shaft, and setting the carrier to the output shaft [13].

### 2.3.2. Shift Mechanism of New Type Transmission

Figure 5 shows the transmission shifting mechanism. The 1st planetary module and 2nd planetary module have same gear components except sun gear. Motor input shaft rotates planetary module's housing and ring gear. If sun gear is fixed planetary gear revolves around sun gear and rotates carrier gear. Gear ratio can be changed by which planetary module's sun gear is fixed. The first gear is a low speed gear; it is used when it takes a lot of load, such as departure and hill driving. When the first gear brake is fixed and the sun gear of the first gear planetary gear set is fixed, the integral planetary gears revolve around the fixed sun gear. At this time, the power is driven to the carrier by the first gear ratio and transmitted to the output shaft. The second gear is a high-speed gear; it is used when vehicle starts and drives at a high-speed. When the second gear brake is fixed and the sun gear of the second gear planetary gear set is fixed, the integral planetary gears revolve around the fixed sun gear. At this time, the power is transferred to the carrier in the second gear ratio and transmitted to the output shaft.

When the sun gear is fixed, the gear ratio of each stage can be calculated through the number of carrier rotations. Table 1 shows the number of revolutions of each part of the first planetary gear module. Table 2 shows the number of revolutions of each part of the second planetary gear module.

**Table 1.** Calculation of 1st planetary gear set gear ratio.

| 1st Gear Ratio | Forward Rotation ($W_1$) | Reverse Rotation ($-W_1$) | $\div (-(1+\frac{S_1}{P_1}\frac{P_2}{R}))$ |
|---|---|---|---|
| $S_1$ | $W_1$ | $0$ | $0$ |
| $P_2$ | $-W_1\frac{S_1}{P_1}$ | $-W_1\left(1+\frac{S_1}{P_1}\right)$ | $W_1\left(\frac{P_1\,R+S_1R}{P_1R+S_1P_1}\right)$ |
| $P_2$ | $-W_1\frac{S_1}{P_1}$ | $-W_1\left(1+\frac{S_1}{P_1}\right)$ | $W_1\left(\frac{P_1\,R+S_1R}{P_1R+S_1P_1}\right)$ |
| C | $0$ | $-W_1$ | $W_1\left(\frac{P_1\,R}{P_1R+S_1P_2}\right)$ |
| R | $-W_1\frac{S_1}{P_1}\frac{P_2}{R}$ | $-W_1\left(1+\frac{S_1}{P_1}\frac{P_2}{R}\right)$ | $W_1$ |

**Table 2.** Calculation of 2nd planetary gear set gear ratio.

| 2nd Gear Ratio | Forward Rotation ($W_1$) | Reverse Rotation ($-W_1$) | $\div (-(1+\frac{S_2}{R}))$ |
|---|---|---|---|
| $S_2$ | $W_1$ | $0$ | $0$ |
| $P_2$ | $-W_1\frac{S_2}{P_2}$ | $-W_1\left(1+\frac{S_2}{P_2}\right)$ | $W_1\left(\frac{P_2\,R+S_2R}{P_2R+P_2S_2}\right)$ |
| C | $0$ | $-W_1$ | $W_1\frac{R}{R+S_2}$ |
| R | $-W_1\frac{S_2}{R}$ | $-W_1\left(1+\frac{S_2}{R}\right)$ | $W_1$ |

$S_1$ is the number of teeth of the 1st gear and $S_2$ is the number of teeth of the 2nd gear. $P_1$ and $P_2$ are the number of teeth of the first and second planetary gears, C is the number of the carrier teeth, and R is the number of teeth of the ring gear. The carrier and the ring gear are shared by the first and second planetary gear sets, so there is no one or two stage separation. Using the calculation results, it is possible to determine the number of teeth of gears satisfying the above ratio, and a transmission prototype design is performed based on this determination [14–18].

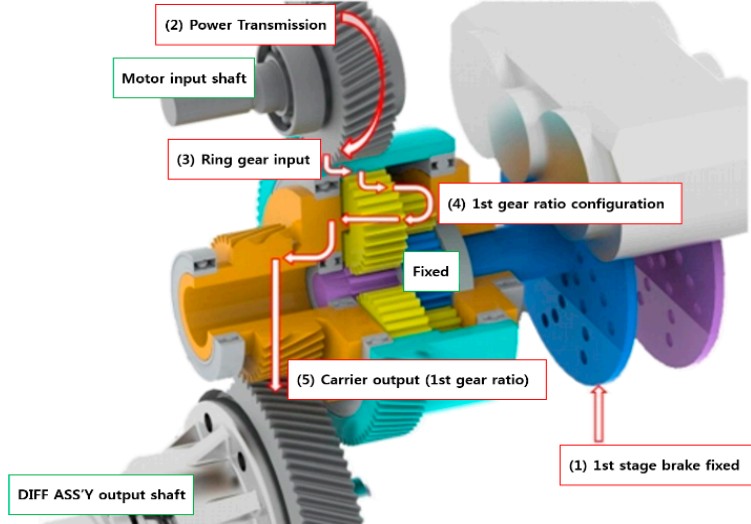

**1st gear shift mechanism**

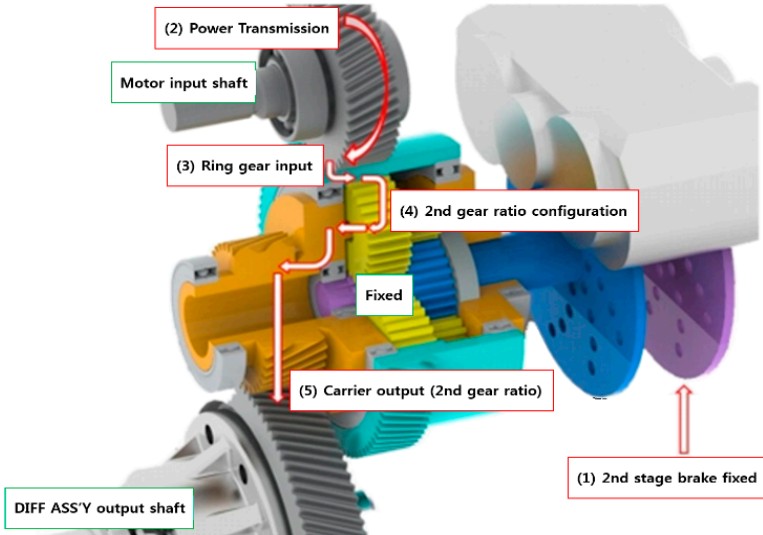

**2nd gear shift mechanism**

**Figure 5.** Shift mechanism of 2-speed transmission for compact electric vehicle using dual disk brake system.

This transmission has two-disc brakes that can selectively lock the two sun-gears of the planetary gear unit. The operation state of the transmission can be changed depending on which combination of the first-stage and second-stage disc brakes is operated. Changes in transmission control according to the presence or absence of operation on the transmission disk brake are summarized in GRAFCET. Figure 6 shows GRAFCET. There are 5 modes: Auto mode, Neutral mode, Manual mode, Parking mode, and Reverse mode. The factors that determine the shift timing of the transmission are the speed of the vehicle and the current gear position. The points at which the number of steps is increased and the points at which the number is decreased must be different from each other so that shifting cannot be performed continuously at a constant speed section. Generally, the motor generates maximum torque from a point where the rotational speed is "0" to a constant speed, the efficiency increases as the rotational speed increases. Therefore, in order to widely use the high efficiency section, it is important to determine the shift point. Basically, the disc brake of the transmission is activated, and power is transmitted via the planetary gear unit to which the Sun-gear is fixed. If both disc brakes do not

operate, the transmission will be in neutral. Conversely, when both disc brakes operate, the power of the motor shaft gear is not transmitted to the differential gear and is changed to the parked state.

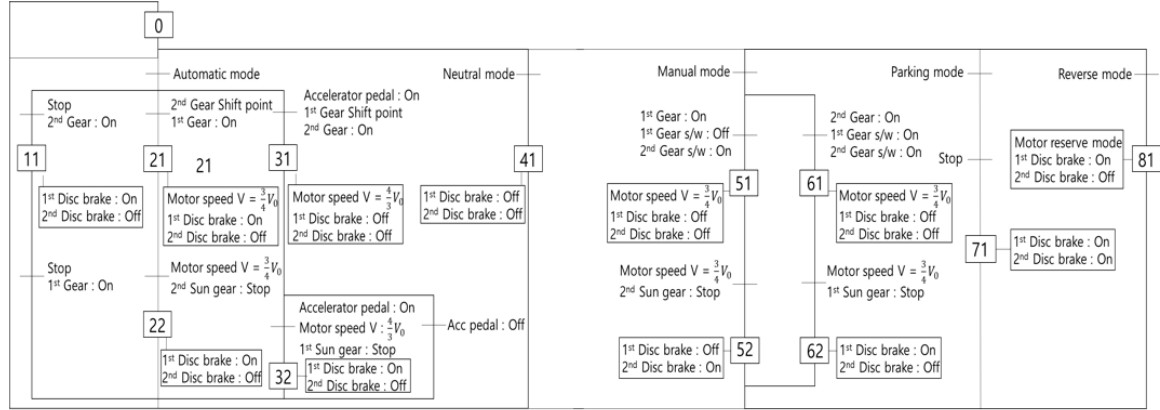

**Figure 6.** 2-speed transmission for compact electric vehicle using dual disc brake control GRAFCET.

## 3. Evaluation of Gears Strength

### 3.1. Calculation of Bending Strength

To obtain the strength of the gear, the following equations are formulas 1 to 4. Equation (1) is the tangential force, Equation (2) is the torque, Equation (3) is the power, and Equation (4) is the main speed. This is commonly used when calculating bending strength and surface durability. To calculate the bending strength of gear teeth, the tangential force must not exceed the gripping pitch circularly permissible tangential force calculated by the allowable root bending stress, and the root shear stress obtained through it shall not exceed the allowable root shear stress. Expressing this as a formula shown in Equation (5). The allowable tangential force can be obtained by using Equation (6), and the root bending stress can be obtained by using Equation (7). The safety factor against the root bending stress is preferably at least 1.2 or more [19–25].

$$F_t = \frac{2000 T_{1,2}}{d_{1,2}} = \frac{1.91 \times 10^7 P}{d_{1,2} \cdot n_{1,2}} = \frac{1000 P}{\gamma} \tag{1}$$

$$T_{1,2} = \frac{F_t \cdot d_{1,2}}{2000} = \frac{1000 P}{\omega_{1,2}} = \frac{9550 P}{n_{1,2}} \tag{2}$$

$$P = \frac{F_t \cdot \gamma}{1000} = \frac{T_{1,2} \cdot \omega_{1,2}}{1000} = \frac{T_{1,2} \cdot n_{1,2}}{9550} \tag{3}$$

$$\gamma = \frac{d_{1,2} \cdot n_{1,2}}{1.91 \times 10^4} \tag{4}$$

$$\sigma_F \leq \sigma_{FP} \tag{5}$$

$$\sigma_F = \frac{F_t}{m_n \cdot b} \left( Y_{FS} \cdot Y_{\varepsilon} \cdot Y_{\beta} \cdot K_A \cdot K_V \cdot K_{F\beta} \right) \tag{6}$$

$$\sigma_{FP} = \frac{1.5 \sigma_{F\,lim} \cdot Y_N \cdot Y_X \cdot B_T}{S_{FM}} \tag{7}$$

### 3.2. Calculation of Surface Durability

To satisfy the surface durability of the gear, the tangential force must not exceed the allowable tangential force of the reference pitch circle calculated by the allowable hertz stress, or the hertz stress obtained from the tangential force should be less than the allowable hertz stress. The equation is shown in Equation (8). The permissible tangential force at the coupling pitch can be obtained by using

Equation (9). The Hertz stress can be obtained using Equation (10). It is preferable that the safety factor against the surface durability has a value of at least 1.1 or more [26–28]

$$F_t \leq F_{t\ lim},\ \sigma_F \leq \sigma_{FP} \tag{8}$$

$$F_{t\ lim} = \sigma_{HP}{}^2 \cdot d_1 \cdot b_H \frac{u}{u \pm 1} \left( \frac{1}{Z_H \cdot Z_C \cdot Z_E \cdot Z_\varepsilon \cdot Z_\beta} \right)^2 \frac{1}{K_A \cdot K'_V \cdot K'_{H\beta} \cdot K'_{Ha}} \tag{9}$$

$$\sigma_H = Z_H \cdot Z_C \cdot Z_E \cdot Z_\varepsilon \cdot Z_\beta \sqrt{\frac{F_t}{d_1 \cdot b_H} \frac{u \pm 1}{u}} \sqrt{K_A \cdot K_V \cdot K_{H\beta} \cdot K_{Ha}} \tag{10}$$

### 3.3. Final Gear Specification

Using the previously calculated gear bending strength and surface durability, the optimum gear specification was designed to meet the gear ratio of the transmission and the drive conditions. The design is the same as Tables 3–6.

**Table 3.** Gear specification 1.

| Calculation Item | Sign | $Z_M$ | $Z_H$ | $Z_{S1}$ | $Z_{P1}$ |
|---|---|---|---|---|---|
| Teeth | Z | 24 | 72 | 30 | 12 |
| Reduction ratio | I | 3 | | 2.041 | |
| | | 9.953 | | | |
| Helix angle | β | 12 | | 12 | |
| Pressure angle (Normal) | $a_n$ | 20 | | 20 | |
| Pressure angle (Transverse) | $a_t$ | 20.410 | | 20.410 | |
| Module (Normal) | $m_n$ | 1.5 | | 1.5 | |
| Module (Transverse) | $m_t$ | 1.533 | | 1.533 | |
| Shift profile (Normal) | $X_n$ | 0.2 | 0.4 | 0 | 0.3 |
| Shift profile (Transverse) | $X_t$ | 0.195 | 0.391 | 0 | 0.293 |

**Table 4.** Gear specification 2.

| Calculation Item | Sign | $Z_{S2}$ | $Z_{P2}$ | $Z_R$ | $Z_C$ | $Z_D$ |
|---|---|---|---|---|---|---|
| Teeth | Z | 12 | 30 | 72 | 56 | 91 |
| Reduction ratio | I | | 1.166 | | 1.625 | |
| | | | 5.687 | | 1.75 | |
| Helix angle | β | | 12 | | 15 | |
| Pressure angle (Normal) | $a_n$ | | 20 | | 20 | |
| Pressure angle (Transverse) | $a_t$ | | 20.410 | | 20.646 | |
| Module (Normal) | $m_n$ | | 1.5 | | 2.5 | |
| Module (Transverse) | $m_t$ | | 1.533 | | 2.588 | |
| Shift profile (Normal) | $X_n$ | 0.3 | 0 | 0.4 | 0.2 | 0.1 |
| Shift profile (Transverse) | $X_t$ | 0.293 | 0 | 0.391 | 0.193 | 0.096 |

**Table 5.** Gear specification 3.

| Surface Durability | | $Z_M$ | $Z_H$ | $Z_C$ | $Z_D$ |
|---|---|---|---|---|---|
| Allowable Hertz stress [N/$mm^2$] | | 1630.00 | | 1630.00 | |
| Area Coefficient | $Z_H$ | 2.343 | | 2.388 | |
| Material constant coefficient | $Z_{ME}$ | 198.800 | | 189.800 | |
| Bite Rate Coefficient | $Z_{EP}$ | 0.806 | | 0.775 | |
| Twist Angle Coefficient | $Z_B$ | 1.000 | | 1.000 | |
| Life Coefficient | $Z_N$ | 1.344 | | 1.344 | |
| Dimension Factor | $Z_X$ | 1.000 | | 1.000 | |
| Lubricant coefficient | $Z_{L12}$ | 0.965 | 0.965 | 0.965 | 0.965 |
| Roughness coefficient | $Z_{R12}$ | 0.946 | 0.946 | 0.946 | 0.946 |
| Lubrication rate factor | $Z_{V12}$ | 0.985 | 0.985 | 0.984 | 0.984 |
| Hardness Ratio Factor | $Z_{W1}, Z_{W2}$ | 1.000 | 1.000 | 1.000 | 1.000 |
| Worst load point | $Z_{C1}, Z_{C2}$ | 1.056 | 0.962 | 1.006 | 0.989 |
| Load distribution coefficient | $K_{HB}$ | 1.200 | | 1.200 | |
| Dynamic Load Coefficient | $K_V$ | 1.250 | | 1.100 | |
| Usage Factor | $K_A$ | 1.250 | | 1.250 | |
| Tangential circle force [N] | $F_t$ | 5371.058 | | 16,537.618 | |
| Tooth surface perpendicular load [N] | | 5798.167 | | 17,741.460 | |
| Axial Right-angle Load [N] | | 2184.140 | | 6423.909 | |
| Input gear Safety Factor | | 1.363 | | 1.887 | |
| Output gear Safety Factor | | 1.496 | | 1.919 | |

**Table 6.** Gear specification 3.

| Bending Stress | | $Z_M$ | $Z_H$ | $Z_C$ | $Z_D$ |
|---|---|---|---|---|---|
| Allowable bending stress [ N/$mm^2$] | | 500.00 | | 500.00 | |
| Tooth profile coefficient | $Y_F$ | 2.352 | 1.524 | 2.186 | 2.164 |
| Stress correction factor | $Y_{SA}$ | 1.697 | 2.415 | 1.803 | 1.824 |
| Load Distribution Factor | $Y_E$ | 0.42 | | 0.67 | |
| Twist Coefficient | $Y_B$ | 0.90 | | 0.88 | |
| Life Coefficient | $Y_N$ | 1.19 | | 1.19 | |
| Fatigue Life Factor | $B_T$ | 0.92 | | 0.92 | |
| Dimension Factor | $Y_X$ | 1.04 | | 1.03 | |
| Dynamic Load Factor | $K_V$ | 1.25 | | 1.25 | |
| Overload Factor | $K_A$ | 1.25 | | 1.25 | |
| Tier distribution coefficient | $K_{FB}$ | 1.10 | | 1.20 | |
| Allowable circle force [N] | S & P | 12,400.8 | 13,454.2 | 16,932.5 | 16,916.8 |
| Tangential circle force [N] | $F_t$ | 5391.399 | | 16,537.618 | |
| Tooth surface perpendicular load [N] | | 5798.167 | | 17,741.460 | |
| Axial Right-Angle Load [N] | | 2133.436 | | 6423.909 | |
| Input gear bending strength Safety Factor | | 2.300 | | 1.024 | |
| Output gear bending strength Safety Factor | | 2.495 | | 1.023 | |

Based on the calculated results, the gear of the planetary gear unit was designed, and the housing was designed. Figure 7 uses the 3D design tool to design the transmission's planetary gear unit and transmission housing. The initial transmission of the research was a structure of a motor shaft, a planetary gear module shaft, and a differential gear shaft (hereinafter, three-shaft) I-type arrangement as shown in Figure 3. In such a structure, the housing is large, long, and contains a large amount of lubricating oil. In addition, there is a problem that it is impossible to lubricate without securing another lubricating device. As the research progressed, the design was changed in a triangular arrangement of three-shaft as shown in Figure 7. This was able to significantly reduce the size of the conventional housing and allowed another lubrication-less lubrication inside the housing.

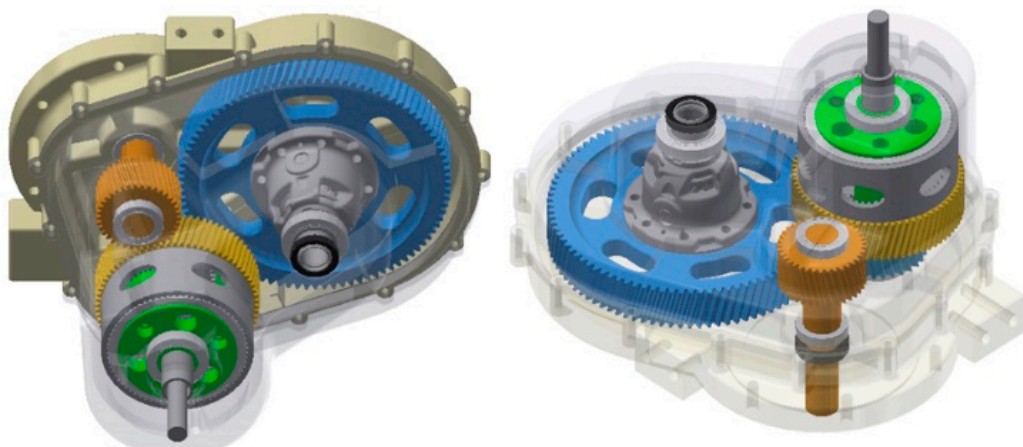

**Figure 7.** 3D drawing of 2nd type transmission.

## 4. Differences Compared to Conventional Transmissions

As shown in Figure 8, to change the structural gear ratio of an existing transmission, the driven shaft and the gear that rotate at different speeds must be connected. For this purpose, a dog clutch is used to connect the power between the gears. Because different rotational speeds cause impacts when the dog-clutch is engaged, the shift quality is not good, and the problem of dock-clutch exhaustion also occurs. Before the dog-clutch is connected to solve this problem, use a friction clutch, called a synchronizer ring, to reduce the speed difference, a synchromesh scheme is used in which connections are made after matching the number of rotations before dog-clutch engagement. However, there is no solution to the root problem of structures that require connections between gears with different rotational speeds listed above [29].

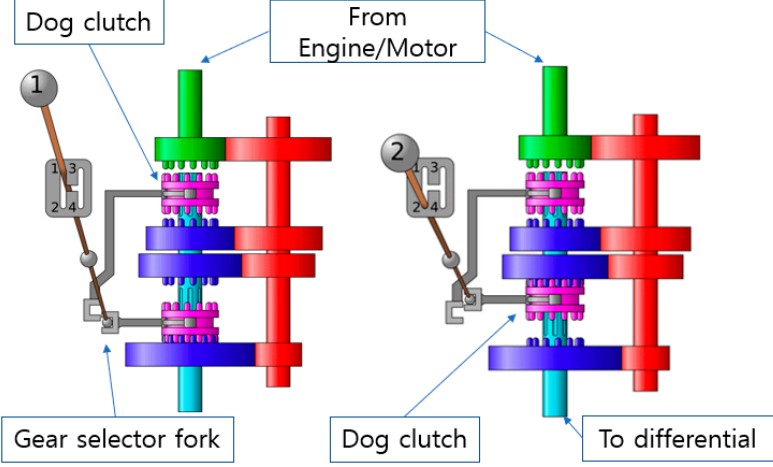

**Figure 8.** Transmission mechanism of existing general transmission.

In the case of the transmission designed in this study, a scheme is used in which the disks of each sun gear are selectively fixed using two sets of planetary gear sets in a different way from the existing transmission. To change the gear ratio, release the disc brake of the existing stage currently connected and put the sun gear of the planetary gear module into a no-load state. When the rotational speed of the motor is synchronized with the next gear shift stage, the rotational speed of the target gear module sun-gear reaches '0'. Finally, the shift is completed when the simple task of securing the target stage disc brake is performed. In the existing transmission, transmission shocks and wear of the transmission elements occur because of the connection between gears with different rotational speeds, whereas in the transmission designed in this study, stopping via motor speed synchronization by using a structure to simply fix the sun gear of the planetary gear module, no shift shock and wear of the shift element occur. Further, in the case of the existing transmission, in the process of synchronizing the speed between the gears, the increase and decrease of the rotational speed of the gear and the consumption of energy accompanying the dog-clutch transfer will occur. The operation only requires energy, that is, it can be determined that the transmission energy efficiency is much higher than that of the conventional transmission, and a limited power source such as a battery is used as an optimum structure for an electric vehicle where efficiency management is important.

## 5. Conclusions

(1) In order to make a compact transmission for an electric vehicle, the interior of the planetary gear unit was designed not to be in a separated parallel structure but in an integrated serial structure.

(2) A new structure in which the shifting is performed is designed by selectively fixing the sun gear of the planetary gear unit.

(3) In the case of the transmission structure, in principle, the speed of the sun gear can be made to be '0' in the shifting process, so that the energy required for the shift shock and the shift can be greatly reduced.

(4) The design of gear teeth suitable for compact electric vehicle transmissions was verified, and the usability of gear trains was verified by surface durability and bending strength.

(5) This transmission was designed with two pairs of planetary gear modules as an integrated structure, and the control system was established by using the GRAFSET.

(6) The planetary gear module was designed based on the calculated gear tooth profile, and the transmission and transmission housing were modeled in 3D.

Through the above research process, the structure of the transmission developed in this paper has reduced the number of parts by sharing ring gear and carrier gear through integration of two sets of planetary gear modules. As the research progresses, the power transmission structure of the 3shaft I type arrangement is replaced with the power transmission structure of the 3shaft triangular arrangement, which makes it easier to lubricate and greatly reduce the size of the housing. The gears used in the transmissions were designed with tooth profile by calculation of surface durability and bending strength. The two-speed transmission for electric vehicles which was developed through the process of this research has significantly reduced the energy required for shift shocks and shifts. Next it will be necessary to verify the theory through a trial production run. It is planned to diversify this technology so that it can be used not only for small electric vehicles, but also for various sizes of vehicles through the diversification of planetary gear units.

**Author Contributions:** Writing-Original Draft Preparation & Writing-Review & Editing—J.-O.H.; Conceptualization and formal analysis—J.-W.S.; Investigation and Reference collection—J.-C.K.; Review—S.-H.O.; The paper was written by all the authors.

**Funding:** This research was supported by the World Class 300 Project (R&D) (S2482370, Development of an integral type transmission system for a Carrier type electric vehicle applying the weight reduction technology and Differential Assembly) of the MOTIE, MSS (Korea). This research was supported by the Chung-Ang University Research Scholarship Grants in 2017*.

**Conflicts of Interest:** The authors declare no conflict of interest.

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
