# Peer review of "Design 2-Speed Transmission for Compact Electric Vehicle Using Dual Brake System"

_applsci, doi:10.3390/app9091793_

Round 1

Reviewer 1 Report

The state of art is not completed. Other similar 2-speed transmissions are not described. It is essential to justify the contribution of this paper.

Page 4:”Figure 5 and it…” is not clear.

The GRAFCET methodology is not introduced.

The GRAFCET diagram in Figure 6 is not explained and it is difficult to understand.

Equations 1 to 7 are not explained.

Tables 3 to 6: standardization of the decimal figures.

Chapter 3: adding discussion about the results.

Conclusions: modifications of the chapter number (it is number 4).

Conclusion 1 is a part of the chapter 2.

Advantages are introduced at the end of the conclusions (“the structure of the transmission developed in this paper has reduced the number of parts”). Nevertheless it is not shown along the paper another transmission to compare with, so it is not possible to demonstrate the previous assertion.

Author Response

1.     The state of art is not completed. Other similar 2-speed transmissions are not described. It is essential to justify the contribution of this paper.

à Added section 4. And compared to the existing transmission and a transmission designed in this study.

2.     Page 4:” Figure 5 and it shows…” is not clear.

à Page 4:” Figure 5 shows …” 

3.     The GRAFCET methodology is not introduced.

 à Added content

4.     The GRAFCET diagram in Figure 6 is not explained and it is difficult to understand.

à Added content

5.     Equations 1 to 7 are not explained.

 à I briefly mentioned the equation. Equations (1) to (4) are the basic formulas required for calculation, based on this, the derived equations are from Equations (5) to (7).

6.     Tables 3 to 6: standardization of the decimal figures.

 à It has been corrected to the third decimal place

7.     Chapter 3: adding discussion about the results.

 à Added content.

8.     Conclusions: modifications of the chapter number (it is number 4).

 à Conclusion The chapter number has been changed.

9.     Conclusion 1 is a part of the chapter 2.

 à I changed the contents of conclusion 1.

10.  Advantages are introduced at the end of the conclusions (“the structure of the transmission developed in this paper has reduced the number of parts”). Nevertheless it is not shown along the paper another transmission to compare with, so it is not possible to demonstrate the previous assertion.

à The conclusions were modified. And added chapter 4 content.

Reviewer 2 Report

-The object of this paper is to design a two-speed transmission which is lighter in weight. But without the comparison between the new design and some original design, readers would not be able to tell how much weight had been saved.

-In the paper, authors did not specify any information of the electric motor which would be paired with this transmission. Without the specifications of the motor, it is difficult to judge whether the new design transmission is suitable for the powertrain system. Readers also have difficult to figure out how this design can save electric energy.

Author Response

1)    The object of this paper is to design a two-speed transmission which is lighter in weight. But without the comparison between the new design and some original design, readers would not be able to tell how much weight had been saved.

à The purpose was to create a two-speed transmission for an electric car that shifts in a new way. While conducting this research, various ideas came out and I was able to reduce the size of the transmission.

à I modified about ‘Lightweight’.

à Related content was created in Chapter 4.

à Related content was created in Chapter 3.3 last paragraph.

Based on the calculated results, the gear of the planetary gear unit was designed, and the housing was designed. Figure 7 uses the 3D design tool to design the transmission's planetary gear unit and transmission housing. The initial transmission of the research was a structure of a motor shaft, a planetary gear module shaft, and a differential gear shaft (hereinafter, three-shaft) I-type arrangement as shown in figure.3. In such a structure, the housing is large, long, and contains a large amount of lubricating oil. In addition, there is a problem that it is impossible to lubricate without securing another lubricating device. As the research progressed, the design was changed in a triangular arrangement of three-shaft as shown in figure.7. This was able to significantly reduce the size of the conventional housing and allowed another lubrication-less lubrication inside the housing.

2-1) In the paper, authors did not specify any information of the electric motor which would be paired with this transmission. Without the specifications of the motor, it is difficult to judge whether the new design transmission is suitable for the powertrain system.

à Related content was created in Chapter 2-1.

2.1. The purpose of the study

Most cities around the world have large and small hills and unpaved roads, and it takes a lot of torque to travel through these roads. In order to cope with such various driving environments, a transmission that optimizes the torque and the speed according to the running speed is required, it is necessary for an electric car with a low output relative to the weight of the vehicle. In this study, we developed a transmission optimized for an electric motor of 88 KW with a maximum torque of 300 Nm when the motor RPM is 2850. A two - speed transmission system was planned to use a planetary gear set and its structure was designed. Also, the gear train was designed based on the calculated gear strength.

2-2) Readers also have difficult to figure out how this design can save electric energy.

à Related content was created in Chapter 4.

4. Differences compared to conventional transmissions

As shown in figure.8, to change the structural gear ratio of an existing transmission, the driven shaft and the gear that rotate at different speeds must be connected. For this purpose, a dog clutch is used to connect the power between the gears. Because different rotational speeds cause impacts when the dog-clutch is engaged, the shift quality is not good, and the problem of dock-clutch exhaustion also occurs. Before the dog-clutch is connected to solve this problem, use a friction clutch, called a synchronizer ring, to reduce the speed difference, a synchromesh scheme is used in which connections are made after matching the number of rotations before dog-clutch engagement. However, there is no solution to the root problem of structures that require connections between gears with different rotational speeds listed above.[30]

 In the case of the transmission designed in this study, a scheme is used in which the disks of each sun gear are selectively fixed using two sets of planetary gear sets in a different way from the existing transmission. To change the gear ratio, release the disc brake of the existing stage currently connected and put the sun gear of the planetary gear module into a no-load state. When the rotational speed of the motor is synchronized with the next gear shift stage, the rotational speed of the target gear module sun-gear reaches '0'. Finally, the shift is completed when the simple task of securing the target stage disc brake is performed. In the existing transmission, transmission shocks and wear of the transmission elements occur because of the connection between gears with different rotational speeds, whereas in the transmission designed in this study, stopping via motor speed synchronization by using a structure to simply fix the sun gear of the planetary gear module, no shift shock and wear of the shift element occur. Further, in the case of the existing transmission, in the process of synchronizing the speed between the gears, the increase and decrease of the rotational speed of the gear and the consumption of energy accompanying the dog-clutch transfer will occur. The operation of only requires energy. That is, it can be determined that the transmission energy efficiency is much higher than that of the conventional transmission, and a limited power source such as a battery is used as an optimum structure for an electric vehicle where efficiency management is important.

Reviewer 3 Report

1) Introduction section needs to be elaborated with more references.

2)  In the abstract, it is said that fuel efficiency can be improved by the proposed transmission system. But it is nowhere mentioned in the manuscript quantitatively how much efficiency is improved?

3) Authors should compare their final design with existing gear system for the electric vehicle in the Tables.

4) After section 3, section 6  (CONCLUSION) is starting. No mention of section 4 and 5?

Author Response

1) Introduction section needs to be elaborated with more references.

à Four references added to the introduction.

1. K.T. Chau, Y.S. Wong, “Overview of Power Management in Hybrid Electric Vehicles”, Energy Conversion and Management 43, p. 1953–1968, 2002

2. C.F. Lin, Y. Chuang,"Energy Management Strategy and Control Laws of An Inverse Differential Gear Hybrid Vehicle", World Electric Vehicle Journal, Vol.4, No.1, pp98-103, 2010

3. G.S. Lee, D.H. Kim, J.H. Han, M.H. Hwang, H.R. Cha,"Optimal Operating Point Determination Method Design for Range-Extended Electric Vehicles Based on Real Driving Tests", Energies, Vol.12, No.5, 845, 2019

4. Toyota “Toyota technical review”, 2010

2)  In the abstract, it is said that fuel efficiency can be improved by the proposed transmission system. But it is nowhere mentioned in the manuscript quantitatively how much efficiency is improved?

à In the contents of Abstract, corrected the contents of fuel efficiency improvement.

3) Authors should compare their final design with existing gear system for the electric vehicle in the Tables.

à Added section 4. And compared to the existing transmission and a transmission designed in this study.

4) After section 3, section 6 (CONCLUSION) is starting. No mention of section 4 and 5?

à Reorganized sections and added section 4. The conclusion is section 5.

Round 2

Reviewer 1 Report

- Resolution improvement in “Figure 6. 2-speed transmission for compact electric vehicle using dual disc brake control GRAFCET”

Author Response

- Resolution improvement in “Figure 6. 2-speed transmission for compact electric vehicle using dual disc brake control GRAFCET”

--> I replaced the image with a good resolution.

Reviewer 3 Report

My concerns are addressed appropriately by the authors. I recommend accepting the manuscript in the present form.

Author Response

Okay, I will do that.

Thank you for your review.
